# Analyses of the Chemical Composition of Plasma-Activated Water and Its Potential Applications for Vaginal Health

**DOI:** 10.3390/biomedicines11123121

**Published:** 2023-11-23

**Authors:** Hyun-Jin Kim, Hyun-A Shin, Woo-Kyung Chung, Ae-Son Om, Areum Jeon, Eun-Kyung Kang, Wen An, Ju-Seop Kang

**Affiliations:** 1Department of Pharmacology, College of Medicine, Hanyang University, Seoul 04736, Republic of Korea; hope0211@hanyang.ac.kr (H.-J.K.); yellowhyun74@hanyang.ac.kr (H.-A.S.); areumii0904@hanyang.ac.kr (A.J.); silverk1239@hanyang.ac.kr (E.-K.K.); anwen@hanyang.ac.kr (W.A.); 2Department of Food and Nutrition, Hanyang University, Seoul 04736, Republic of Korea; entksv10@naver.com (W.-K.C.); aesonom@hanyang.ac.kr (A.-S.O.)

**Keywords:** plasma-activated water (PAW), hypochlorous acid (HOCl), probiotics (*Lactobacillus reuteri*), mucosa protection, vaginal cleansing effect

## Abstract

This study aimed to elucidate the unique chemical compositions of plasma-activated water (PAW) and the potential antibacterial efficacy of PAW as a novel vaginal cleanser. We analyzed the ion compositions (four anions: F^−^, Cl^−^, NO_3_^−^, SO_4_^2−^; five cations: Na^+^, NH_4_^+^, K^+^, Mg^2+^, Ca^2+^) of several formulations of PAW generated at different electrical powers (12 and 24 V) at various treatment time points (1, 10, and 20 min), and stay durations (immediate, 30, and 60 min). As treatment duration increased, hypochlorous acid (HOCl), Ca^2+^, and Mg^2+^ concentrations increased and Cl^−^ concentration decreased. Higher electrical power and longer treatment duration resulted in increased HOCl levels, which acts to prevent the growth of general microorganisms. Notably, PAW had no antibacterial effects against the probiotic, *Lactobacillus reuteri*, which produces lactic acid and is important for vaginal health. These findings indicate that PAW contains HOCl and some cations (Ca^2+^ and Mg^2+^), which should help protect against pathogens of the vaginal mucosa and have a cleansing effect within the vaginal environment while not harming beneficial bacteria.

## 1. Introduction

Plasma technology has been proven to be effective in various medical applications, including regenerative medicine for skin and dental treatments, as well as surface sanitization and sterilization of medical tools [1]. Plasma is a term that refers to a quasi-neutral ionized gas containing photons, free radicals, and ions as well as uncharged particles [2,3,4]. Traditional plasmas are categorized as thermal or non-thermal, based on the thermodynamic equilibria and non-equilibrium of electrons and other gas species. Recent advances in plasma engineering have enabled the generation of plasma-activated waters (PAW) through non-thermal atmospheric pressure plasmas (NTAPPs) [5]. The PAW can be produced by exposing water to ionized gas generated by a plasma device, either above or below the water surface. Many studies have demonstrated that the reactive species in PAW are not toxic and do not pollute the environment, paving the way for diverse applications of PAW in the life sciences [4]. The primary application areas of PAW technology are seed germination, plant growth, food preservation, antimicrobial activities, virus inactivation, and anticancer treatments, among others [6]. The disinfection effectiveness of PAW hinges upon the concentration of reactive oxygen species (ROS) and reactive nitrogen species (RNS) in the PAW. ROS generated in PAW, including hydrogen peroxide(H_2_O_2_), hydroxyl radical(∙OH), and ozone (O_3_), function as potent oxidizing agents, inducing oxidative stress on the microbial cell membranes and, consequently, bacterial damage and death [7]. RNS present in PAW, such as nitric oxide (NO), nitrite (NO_2_^−^), and nitrate (NO_3_^−^), exist primarily as peroxynitrite (ONOOH) under acidic conditions. Peroxynitrite can accumulate inside cells leading to apoptotic or necrotic cell death [8]. The production of various reactive oxygen and nitrogen species (RONS) depends on the operating parameters of plasma generation, such as the power source, treatment time, feed gases, and electrode materials [7,9]. Gao et al. reported that increasing the electrical power from 0 to 160 W led to higher concentrations of OH, H_2_O_2_, NO_3_^−^, and NH_4_ in PAW [4]. Berardinelli et al. reported that longer plasma treatment duration result in elevated concentrations of NO_2_^−^ and NO_3_^−^ in PAW [10]. Other studies have demonstrated that the concentrations of O_3_ and H_2_O_2_ can also be increased in PAW by manipulating how the PAW is generated [11]. Georgescu et al. reported that feed gases like helium and nitrogen can alter the reactive species in PAW by changing the electron density and surface charges compared with ambient air [12]. These parameters can be fine-tuned to create PAWs with different reactivity levels for various disinfection applications, and researchers are actively developing PAW devices to generate PAWs with different reactivities [13]. The applications of PAW have rapidly expanded to include the treatment of biomedical devices and biological materials, including foods.

Lee and Hong demonstrated that plasma discharge in tap water helps eliminate harmful microorganisms by increasing the concentrations of free residual chlorine molecules such as hypochlorous acid and hypochlorite ions [14,15,16]. While the exact mechanisms remain unclear, PAW has been shown to have antibacterial and cytotoxic activity, which has been attributed to the reactive species present in PAW [4,5,17,18]. Further theoretical-experimental investigations are warranted to expand our understanding of PAW and further explore its potential in biological decontamination and clinical applications [12].

In previous studies, we confirmed the bactericidal effect of PAWs on clinically abnormal vaginal microbiota in clinical practice [14,19,20]. Patients sprayed with PAW (22.3%) had fewer Gram-positive and -negative bacteria than betadine treatment (BT) patients (14.4%) [14]. A significant decrease after treatment was observed in the following pathogenic organisms: *Mycoplasma hominis* (30 ± 15.28% decrease), *Ureaplasma urealyticum* (25 ± 9.93% decrease), *Ureaplasma parvum* (23 ± 8.42% decrease), and *Candida albicans* (28 ± 10.86% decrease) [19]. Vaginitis is a common disease among women and bacterial vaginosis is the most common form of vaginitis, accounting for 22~50% of all vaginitis cases, followed by candida vulvovaginitis at 17~35%, and trichomonas vaginitis at 4~39%. Vaginosis is caused by unbalanced changes in the vaginal microbiome, which are associated with a reduction in the overall number of Lactobacilli species and a predominance of anaerobic microorganisms, including *Gardnerella vaginalis*, *Trichomonas*, and *Candida albicans* strains. Generally, the clinical symptoms include a foul-smelling vaginal discharge, fever, sexual discomfort and painful urination [21].

The excessive growth of anaerobic species, particularly *Gardnerella vaginalis*, results in a polymicrobial biofilm that adheres to the vaginal epithelium [22]. Biofilms are communities of microorganisms encased in a polymeric matrix of nucleic acids, polysaccharides, and proteins [23]. Biofilm-related infections are challenging to eradicate by both the immune system and antibiotics, leading to a high rate of relapse and recurrence in bacterial vaginosis cases [24,25,26].

PAW can be used to disrupt biofilms and effectively eliminate the attached bacteria. PAW created through cold atmospheric-pressure plasma discharge in water demonstrated significant antimicrobial activity against biofilms, without promoting bacterial resistance [27].

Therefore, our objective in this study was to analyze the chemical components underlying the disinfection effects of PAW with a specific focus on vaginal sterilization. Additionally, we evaluated the effects of PAW on *Lactobacillus reuteri*, a vital probiotic component of the vaginal microbiome essential for the host’s health [28].

## 2. Materials and Methods

### 2.1. PAW Generating System and PAW Processing

The PAW system comprised an underwater plasma-generating device, with plasma generated using procedures established previously [20]. An illustration of the device is provided in Figure 1; the device comprises a 3 L cleaning solution container, the atmospheric plasma electrodes, and a controller.

The device facilitates plasma discharge underwater through a plasma electrode located at the container’s base. The plasma generation modules consist of two plasma electrodes separated by an insulating frame, with each electrode connected to power of different polarity. These electrodes are disc-shaped grids, measuring 77 mm in diameter and 0.5 mm in thickness, with titanium used as the electrode material to prevent corrosion. The grid has a diameter of 1.07 mm and pitch of and 3.92, resulting in 50% open space. To enhance plasma generation, a 300 nm thick platinum thin film is deposited on the grid’s surface using electroless plating. Plasma discharge occurs between the two electrodes with a consistent 2 mm separation maintained by the insulating frame (Figure 1B) [20]. To analyze the chemical composition of water at different positions (top, middle, bottom) within the 3 L cleaning solution container, three stopcocks were connected to the container (Figure 1A). Figure 1C shows the operation process of the plasma device.

### 2.2. PAW Sample Preparation Conditions

PAW samples were prepared using the following conditions: (1) electrical power of 12 or 24 V; (2) plasma treatment durations of 1, 10, or 20 min; and (3) 0, 30 or 60 min retention times after plasma exposure (Table 1). At specified time intervals, 10 mL samples of PAW were collected and immediately subjected to analysis using ion chromatography, and a residual chlorine analyzer.

### 2.3. Analytical Methods

The inorganic constituents and hypochlorous acid content of the PAW were determined as described below.

#### 2.3.1. Ion Chromatography (IC)

IC analysis was carried out with an ion chromatograph (Dionex ICS-3000, Thermo Fisher Scientific, Waltham, MA, USA) equipped with both an anion and cation module. An Ionpac AG20 4 × 50 mm guard column (Thermo Fisher Scientific, Waltham, MA, USA) and Ionpac AS20 4 × 250 mm analytical column (Thermo Fisher Scientific, Waltham, MA, USA) were employed in the anion module while an Ionpac CG16 5 × 50 mm guard column and Ionpac CS 5 × 250 mm analytical column were used in the cation module. The column temperature was set to 30 °C for a run time of 20 min. A gradient method was used for the mobile phase of the anion module starting at 12 mM sodium hydroxide (NaOH) for the first 8 min, followed by a change to 40 mM NaOH from 8 min to 12 min, maintenance at 40 mM NaOH until 18 min, and then a decrease to 12 mM NaOH until 20 min. An isocratic method was employed in the cation module with a mobile phase consisting of 40 mM methanesulfonic acid (MSA). The flow rate was 1 mL/min and the injection volume was 25 μL. The anion module contained an ADRS 600 suppressor (Thermo Fisher Scientific, Waltham, MA, USA) while the cation module incorporated a CDRS 600 suppressor (Thermo Fisher Scientific, Waltham, MA, USA). Instrument control and data acquisition were managed through Chromeleon^®^ chromatography management software (version 6.80) (Thermo Fisher Scientific, Waltham, MA, USA).

Working standards for seven anions were prepared (fluoride, chloride, nitrite, bromide, nitrate, phosphate, sulfate) and their calibration curves ranges were 0.1 to 9.9, 0.15 to 15, 0.5 to 50, 0.5 to 50, 0.5 to 50, 0.75 to 75, and 0.75 to 75 mg/L, respectively.

Cation standard working solutions of six cations were prepared (lithium, sodium, ammonium, magnesium, potassium, calcium) and their calibration curves ranges were 0.3 to 25, 1 to 100, 1.26 to 126, 1.27 to 127, 2.5 to 250, and 2.5 to 250 mg/L, respectively.

#### 2.3.2. Residual Chlorine Analyzer

Hypochlorous acid concentrations in the PAW samples were determined using DPD(N,N-diethyl-1,4-phenylenediamine) free chlorine reagent with a Q-CL501B analyzer (Shenzhen Sinsche Technology Co. Ltd., Shenzhen, China), following the DPD colorimetric method.

### 2.4. Evaluation of Antibacterial Efficacy

#### 2.4.1. Microorganisms and Materials

To assess antibacterial activity, *Limosilactobacillus reuteri* subsp. *reuteri* (*L. reuteri*) was sourced from the Korean Collection for Type Cultures (KCTC) (Table 2). To cultivate *L. reuteri* subsp. *reuteri* (*L. reuteri*), de Man, Rogosa and Sharpe agar (MRS, Difco, Detroit, MI, USA) was employed as the growth medium.

#### 2.4.2. Antibacterial Activity

The antibacterial activity of plasma-activated water (PAW) was determined using the filter paper disc method [29]. Bacterial cultures (sub-cultured before assay) were diluted with sterile water to obtain a bacterial suspension of OD_600nm_ = 0.2~0.3. Petri dishes containing 10 mL MRS media were inoculated with 0.1 mL of the bacterial suspension, dried within a sterile chamber, and incubated at 37 °C for 48 h. A filter paper disc (Ø: 6 mm, ADVANTEC, Tokyo, Japan) was impregnated with 20 μL PAW sample, placed on the medium, and then the petri dish was left at 37 °C for 15 min or 30 min. Sterile water served as the negative control. All experiments were performed in triplicate.

## 3. Results

### 3.1. Inorganic Anion and Cation Composition of PAW Samples

The inorganic anion and cation content of PAW samples generated by underwater plasma discharge for various durations (1, 10, and 20 min) at two voltage settings (12 and 24 V) was determined. Chromatography results for the standard anions and cations are shown in Figure 2. Retention times for the individual anions (fluoride, chloride, nitrite, bromide, nitrate, phosphate, sulfate) were approximately 3.74, 5.14, 6.05, 7.06, 7.87, 9.52, and 14.30 min, respectively. The retention times for the cations (lithium, sodium, ammonium, magnesium, potassium, calcium) were 4.78, 6.45, 7.94, 10.84, 11.96, and 14.77 min, respectively. The calibration curves for the standard anions and cations demonstrated linearity, indicating that anion and cation concentrations in the PAW samples could be determined accurately. Chromatographs of anions and cations within the PAW samples are shown in Figure 2C,D.

Ion chromatography analysis revealed the generation of five cations and four anions following plasma treatment of tap water (Table 3 and Table 4). Table 3 presents changes in ion concentrations following 12 V plasma generation. After plasma activation for 20 min, changes in the concentrations of Ca^2+^, Mg^2+^, and Cl^−^ were noted. The baseline levels of Ca^2+^ were 10.46 mg/L (top), 15.17 mg/L (middle), and 15.00 mg/L (bottom), while those of Mg^2+^ were 3.52 mg/L (top), 3.86 mg/L (middle), and 3.89 mg/L (bottom). The baseline levels of Cl^−^ were 20.21 mg/L (top), 19.35 mg/L (middle), and 19.27 mg/L (bottom). The Ca^2+^ concentrations were 20.73 mg/L (top), 19.33 mg/L (middle), and 18.75 mg/L (bottom) after a 60 min treatment duration, while those of Mg^2+^ were and 4.51 mg/L (top), 4.04 mg/L (middle), and 4.01 mg/L (bottom). The Cl^−^ concentrations were 19.76 mg/L (top), 18.46 mg/L (middle), and 17.98 mg/L (bottom) under the same conditions. After plasma activation for 10 min, changes were observed in Ca^2+^ and Cl^−^. Ca^2+^, initially at 13.38 mg/L, changed to 17.33 (top), 17.76 (middle), 17.73 mg/L (bottom) after a 60 min treatment duration. Cl^−^ concentration, initially at 17.41 mg/L, changed to 16.85 (top), 16.96 (middle), and 17.17 mg/L (bottom) under the same conditions. After plasma activation for 1 min, changes were observed in Ca^2+^ and Cl^−^. The baseline levels of Ca^2+^ and Cl^−^ ions were 13.31 mg/L and 19.08 mg/L. The Ca^2+^ concentrations were 16.33 (top), 16.20 (middle), and 16.31 mg/L (bottom) for a treatment time of 60 min. The Cl^−^ concentrations were 18.62 (top), 13.83 (middle), and 13.62 mg/L (bottom) under the same conditions.

Table 4 presents changes in ion concentrations after 24 V plasma generation. After plasma activation for 20 min, changes were observed in Ca^2+^ and Cl^−^. The baseline levels of Ca^2+^ were 14.61 mg/L (top), 14.35 mg/L (middle), and 12.17 mg/L (bottom), while those of Cl^−^ were 18.25 mg/L (top), 17.93 mg/L (middle), and 17.10 mg/L (bottom). The Ca^2+^ concentrations reached 16.78 mg/L (top), 16.79 mg/L (middle), and 15.95 mg/L (bottom) after a 60 min treatment duration. The Cl^−^ concentrations reached 16.17 mg/L (top), 15.55 mg/L (middle), and 15.28 mg/L (bottom) under the same conditions. After plasma activation for 10 min, the baseline concentration of Ca^2+^, initially at 13.31 mg/L, increased to 16.98 (top), 16.07 (middle), and 16.08 mg/L (bottom) within 60 min. Cl^−^, initially at 19.08 mg/L, decreased to 17.35 (top), 17.18 (middle), and 17.15 mg/L (bottom) under the same conditions. After plasma activation for 1 min, the baseline concentration of Ca^2+^, initially at 12.43 mg/L, increased to 16.42 (top), 13.80 (middle), and 16.54 mg/L (bottom) after a 60 min treatment duration. To summarize, there was an increase in Ca^2+^ concentration and a decrease in Cl^−^ concentration with plasma generation. Moreover, as the duration of plasma generation increased, the magnitude of these ion changes became more pronounced. However, there were no significant discernible effects of electrical power or retention time after plasma exposure on ion concentrations.

### 3.2. Hypochlorous Acid (HOCl) Concentration in PAW

Chlorine is routinely added to tap water to inhibit the growth of microorganisms, including common bacteria and *E. coli*, during the tap water supply process. As a result, residual chlorine is present in tap water in both free and combined forms. Free residual chlorine encompasses species such as HOCl, OCl^−^, and Cl^−^, with the chemical equation indicating the formation of hydrochloric acid and hypochlorous acid as follows:Cl_2_ + H_2_O ⇌ HOCl + HCl(1)

Hypochlorous acid is inherently unstable and may dissociate into the hypochlorite anion:HOCl ⇌ ClO^−^ + H^+^(2)

The presence of HOCl or OCl^−^ depends on the acidity (pH 4–6) or basicity (pH 8.5–10) of the water, with HOCl being formed under acidic condition and OCl^−^ being formed under basic conditions. Hypochlorous acid concentrations increased significantly following plasma treatment of tap water, and these concentrations were maintained for 60 min (Table 3 and Table 4).

Figure 3 depicts the changes in hypochlorous acid concentration after generation of 12 and 24 V plasma. In Figure 3A, the concentrations of hypochlorous acid immediately after treatment with 24 V plasma for 20 min were 2.10 mg/L (top), 2.64 mg/L (middle), and 2.00 mg/L (bottom). For 12 V plasma, the immediately generated hypochlorous acid concentrations were 1.03 mg/L (top), 1.21 mg/L (middle), and 1.07 mg/L (bottom). As shown in Figure 3B), hypochlorous acid concentrations immediately after treatment with 24 V plasma for 10 min were 1.54 mg/L (top), 1.49 mg/L (middle), and 1.03 mg/L (bottom). For 12 V plasma, the immediately generated hypochlorous acid concentrations were 0.78 mg/L (top), 0.71 mg/L (middle), and 0.59 mg/L (bottom). Figure 3C reveals that hypochlorous acid concentrations generated after treatment with 24 V plasma or 12 V for 1 min were not significantly different. In summary, the amount of hypochlorous acid generated after plasma activation was influenced by electrical power and plasma processing time, but not retention time after plasma exposure.

### 3.3. Antibacterial Activity of PAW

The antibacterial activity of PAW against the Gram-positive bacterium *L. reuteri* was assessed using the filter paper disc method. PAW samples were collected immediately from the middle position after treatment with 12 V plasma for 20 min. Subsequently, *L. reuteri* cultures were exposed to filter paper treated with PAW samples for 15 min or 30 min. PAW had no significant antibacterial effect on *L. reuteri* after either a 15 min (Figure 4A) or a 30 min (Figure 4B) incubation period.

## 4. Discussion

Our innovative vaginal cleaning device employing underwater plasma discharge generates various reactive radicals in tap water, conferring it with antibacterial activity [14,19,20]. The effectiveness of PAW in terms of antibacterial activity is contingent on its chemical composition. Thus, chemical composition analysis was undertaken to elucidate the antibacterial properties of various PAW samples.

Chemical analysis revealed an increase in Ca^2+^ and Mg^2+^ and a decrease in Cl^−^ as the duration of plasma generation increased. The concentration of hypochlorous acid was notably enhanced by higher electrical power input and longer plasma processing times. Previous studies have reported that magnesium and calcium ions have beneficial effects on the epidermis: Mg enhances epidermal barrier function and exhibits anti-inflammatory properties [30,31], while Ca promotes epidermal differentiation and regulates hyaluronic acid synthesis in the epidermal layer [32,33,34,35]. Various ions are constituents of the natural moisturizing factor (NMF) in the stratum corneum, the outermost layer of skin. Ionized minerals strengthen the epidermal barrier, particularly in damaged skin, and hinder the penetration of various external irritants [36]. Therefore, the generation of ions, specifically Ca^2+^ and Mg^2+^, in PAW through plasma treatment suggests that the PAW is likely to have antibacterial and skin protection effects.

Furthermore, free chlorine, including HOCl, is a crucial component of disinfectants. Reactive chlorine species, such as HOCl, are widely employed for disinfection in industrial, hospital, and household settings [37]. As illustrated in Figure 5A, the body’s innate immune system plays a pivotal role in the production of substantial amounts of oxidants, with HOCl being a key component. These oxidants, including HOCl, are generated by neutrophils as a response to the presence of invading pathogens [38]. As depicted in Figure 5B, the HOCl produced possesses a remarkable ability to breach the protective barriers of bacterial cells; HOCl launches rapid and destructive attacks. These attacks result in a cascade of effects within the bacterial cell, including the loss of adenosine triphosphate (ATP), disruption of DNA replication, and inhibition of protein synthesis [37]. This two-step process highlights the crucial role of HOCl in the body’s immune defense mechanisms. HOCl is produced by neutrophils in response to infection and is followed by the infiltration of bacterial cells and the initiation of multiple processes that lead to the destruction of invading pathogens. The increased Ca^2+^ and Mg^2+^ concentrations in PAW generated using our plasma device and the augmentation of hypochlorous acid production indicate that the PAW samples produced using this device are likely to have antibacterial efficacy.

Importantly, PAW did not exhibit antibacterial effects against *L. reuteri*, a probiotic crucial for vaginal health that produces lactic acid and helps maintain an acidic environment in the vagina to inhibit the growth of pathogens [39].

## 5. Conclusions

In this study, we conducted a comprehensive analysis of the chemical composition of PAW to determine the potential applications of PAW for vaginal sterilization and mucosal protection. The major findings of this research can be summarized as follows: There were notable changes in ion composition within PAW following plasma treatment. Specifically, levels of Ca^2+^ and Mg^2+^ increased while those of Cl^−^ decreased, with these changes becoming more pronounced over longer plasma generation times. This suggests that PAW generated by underwater plasma discharge can have different ion compositions depending on the conditions under which the PAW is generated. We found a significant increase in hypochlorous acid within PAW samples immediately after plasma treatment. Moreover, this increased level of hypochlorous acid was sustained for at least 60 min, indicating the potential disinfection capabilities of PAW in comparison with untreated water. Through in-depth exploration of the ion content of PAW, we demonstrated the presence of HOCl, a key component of the body’s innate immune defense mechanisms. HOCl is known for its ability to eliminate invading pathogens efficiently. Intriguingly, PAW had no significant antibacterial effects against *L. reuteri*, a probiotic that plays a crucial role in maintaining vaginal health by producing lactic acid, and thereby creating an acidic environment that hinders pathogen growth. This selective antibacterial activity of PAW is encouraging for its potential use in female intimate hygiene products.

In summary, by providing insights into the ion composition of PAW, we have opened avenues for further exploration and development of feminine hygiene solutions that can harness these unique characteristics while respecting the delicate balance of the vaginal environment. The potential of PAW to offer both protection and cleansing within this context represents a promising direction for future research and applications.

## Figures and Tables

**Figure 1 biomedicines-11-03121-f001:**
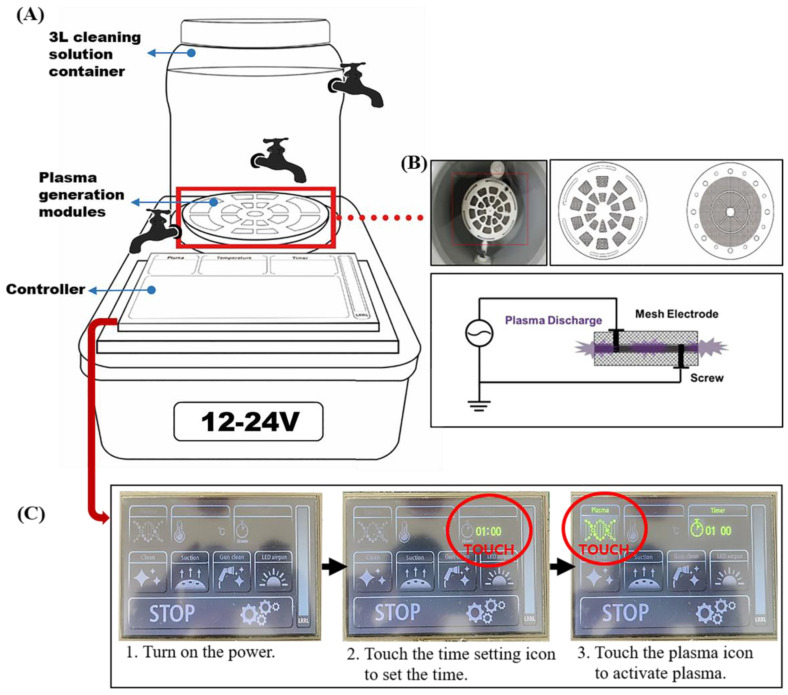
Plasma device and operation process. (**A**) Components of the plasma device used in the experiment. (**B**) Plasma module and schematic principle for underwater plasma discharge. Reprinted with permission from Hwang, et al., 2020 [20], and (**C**) the digital display controller and details of how to operate the plasma device.

**Figure 2 biomedicines-11-03121-f002:**
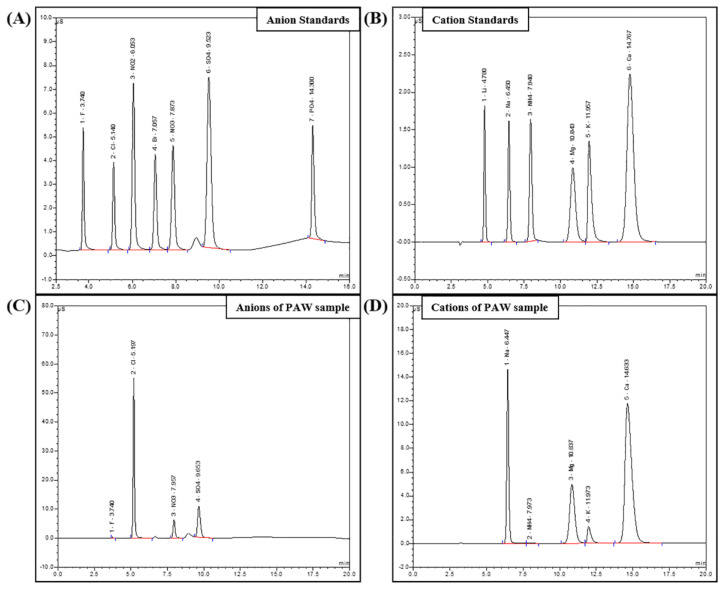
Ion chromatography analysis of inorganic anions and cations in PAW. Chromatograms depict the separation of inorganic anions (**A**) and cations (**B**). Determination of inorganic anions (**C**) and cations (**D**) in PAW generated using 12 V electrical power, 20 min treatment time, sample position at the top, and a 60 min retention time.

**Figure 3 biomedicines-11-03121-f003:**
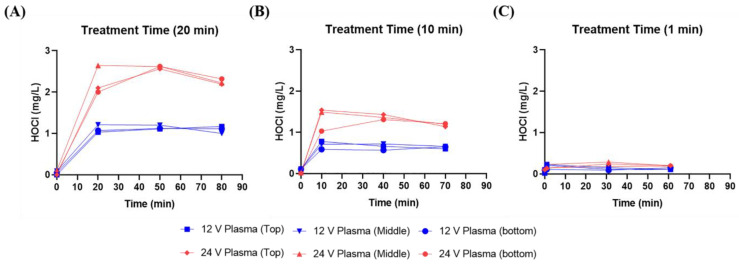
Variation in hypochlorous acid (HOCl) concentrations following plasma discharge for different durations ((**A**): 20 min, (**B**): 10 min, and (**C**): 1 min) at 12 and 24 V power.

**Figure 4 biomedicines-11-03121-f004:**
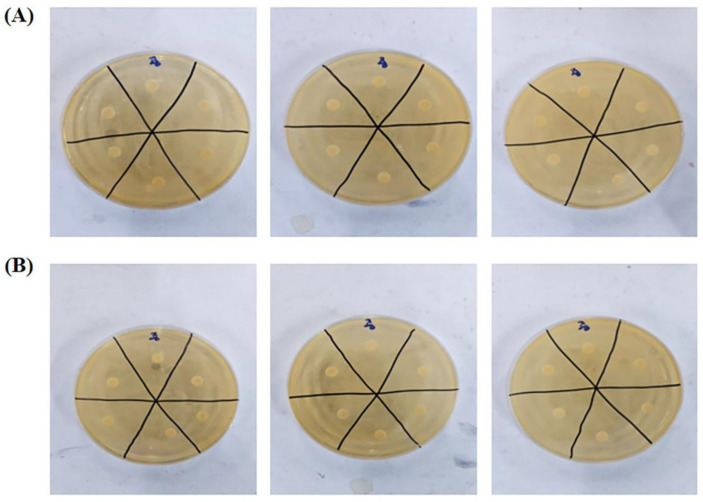
Antibacterial activity of PAW against Gram-positive *L. reuteri* following 15 min (**A**) or 30 min (**B**) exposure. Non-marked: PAW samples. Marked: negative control (sterile water).

**Figure 5 biomedicines-11-03121-f005:**
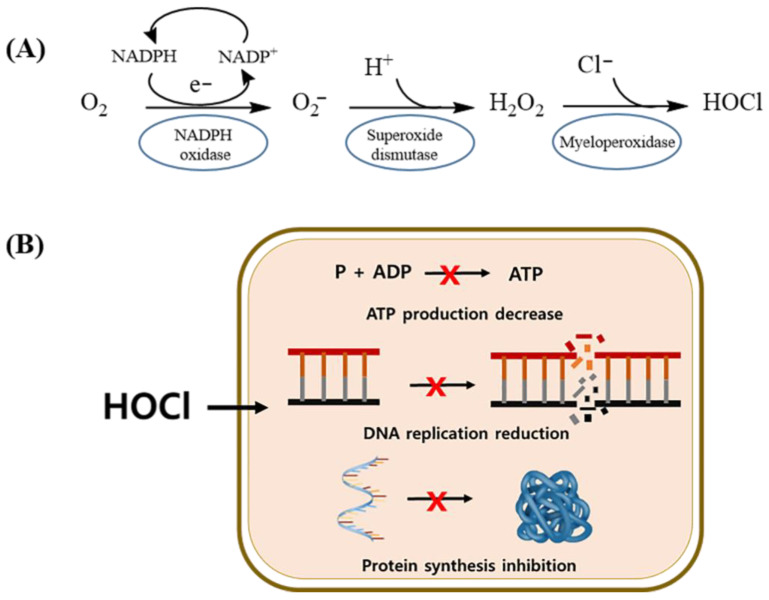
Schematic of HOCl production mechanisms in neutrophils (**A**) and HOCl targets in Gram-negative bacterial cells (**B**).

**Table 1 biomedicines-11-03121-t001:** The conditions of plasma activated water.

	Position	Power	Treatment Time (min)	Retention Time (min)
Tap water	Top, middle, bottom	12 V, 24 V	1, 10, or 20	0, 30, or 60

**Table 2 biomedicines-11-03121-t002:** Microorganisms used for antibacterial activity test.

Microorganism		Strain
Gram-positive	*Lactobacillus reuteri*	KCTC 3594

**Table 3 biomedicines-11-03121-t003:** Measurements of inorganic ions and hypochlorous acid for 12 V underwater plasma discharge.

Sample	Treatment Time	Sampling Position	Retention Time	Na^+^	NH_4_^+^	K^+^	Mg^2+^	Ca^2+^	F^−^	Cl^−^	NO_3_^−^	SO_4_^2−^	HOCl
No.	(min)		(min)	(mg/L)	(mg/L)	(mg/L)	(mg/L)	(mg/L)	(mg/L)	(mg/L)	(mg/L)	(mg/L)	(mg/L)
1	20 min		baseline	9.20	0.18	2.48	3.52	10.46	0.05	20.21	5.79	10.23	0.00
2		Sample Top	0	9.11	0.13	2.43	4.41	18.30	0.06	19.24	5.70	10.25	1.03
3			30	9.16	0.16	2.43	4.48	20.08	0.06	19.96	5.97	10.91	1.11
4			60	9.16	0.13	2.44	4.51	20.73	0.06	19.79	5.82	10.50	1.17
5			baseline	7.46	0.15	2.33	3.86	15.17	0.05	19.35	6.46	9.59	0.09
6		Sample Middle	0	7.45	0.16	2.32	4.02	18.37	0.05	18.54	6.44	9.67	1.21
7			30	7.45	0.18	2.32	4.05	19.15	0.05	18.18	6.31	9.36	1.20
8			60	7.41	0.17	2.31	4.04	19.33	0.05	18.46	6.39	9.53	1.00
9			baseline	7.34	0.14	2.28	3.89	15.00	0.06	19.27	6.96	10.06	0.05
10		Sample Bottom	0	7.06	0.17	2.20	3.84	16.99	0.05	17.31	6.54	9.25	1.07
11			30	7.28	0.16	2.27	3.99	18.40	0.05	17.90	6.77	9.58	1.12
12			60	7.30	0.17	2.27	4.01	18.75	0.05	17.98	6.78	9.63	1.11
13	10 min		baseline	7.59	0.16	2.27	3.40	13.38	0.04	17.41	7.71	9.02	0.11
14		Sample Top	0	7.59	0.22	2.27	3.50	16.22	0.05	17.16	7.75	9.13	0.78
15			30	7.62	0.20	2.28	3.51	17.02	0.05	17.23	7.74	9.09	0.66
16			60	7.57	0.26	2.30	3.50	17.33	0.05	16.85	7.65	8.95	0.60
17		Sample Middle	0	7.65	0.20	2.28	3.55	17.66	0.05	16.92	7.69	8.98	0.71
18			30	7.66	0.20	2.30	3.56	17.74	0.05	16.98	7.63	8.95	0.72
19			60	7.60	0.17	2.28	3.54	17.76	0.05	16.96	7.63	8.96	0.66
20		Sample Bottom	0	7.69	0.17	2.30	3.52	17.71	0.05	17.05	7.57	8.88	0.59
21			30	7.72	0.18	2.29	3.53	17.77	0.05	16.99	7.64	9.00	0.57
22			60	7.67	0.17	2.28	3.51	17.73	0.05	17.17	7.64	9.00	0.65
23	1 min		baseline	7.49	0.23	2.28	3.44	13.31	0.05	19.08	7.84	9.37	0.02
24		Sample Top	0	7.21	0.25	2.05	3.18	16.41	0.06	18.39	7.48	8.80	0.21
25			30	7.20	0.22	2.16	3.18	16.32	0.06	18.83	7.67	9.12	0.11
26			60	7.19	0.19	2.15	3.18	16.33	0.06	18.62	7.60	8.98	0.11
27		Sample Middle	0	6.37	0.08	1.89	3.14	15.03	0.04	14.00	7.44	7.85	0.23
28			30	6.74	0.11	1.94	3.14	15.70	0.05	13.77	7.58	8.59	0.15
29			60	6.42	0.09	1.99	3.22	16.20	0.05	13.83	7.65	7.77	0.11
30		Sample Bottom	0	6.31	0.09	1.92	3.15	16.06	0.05	13.84	7.65	7.90	0.11
31			30	6.26	0.09	1.93	3.15	16.02	0.05	13.76	7.66	7.88	0.09
32			60	6.30	0.11	1.94	3.18	16.31	0.05	13.62	7.61	7.76	0.15

**Table 4 biomedicines-11-03121-t004:** Measurements of inorganic ions and hypochlorous acid for 24 V underwater plasma discharge.

Sample	Treatment Time	Sampling Position	Retention Time	Na^+^	NH4^+^	K^+^	Mg^2+^	Ca^2+^	F^−^	Cl^−^	NO3^−^	SO4^2−^	HOCl
No.	(min)		(min)	(mg/L)	(mg/L)	(mg/L)	(mg/L)	(mg/L)	(mg/L)	(mg/L)	(mg/L)	(mg/L)	(mg/L)
1	20 min		baseline	6.92	0.17	2.22	3.33	14.61	0.05	18.25	7.34	8.66	0.00
2		Sample Top	0	6.95	0.15	2.22	3.45	16.12	0.04	15.46	7.20	8.37	2.10
3			30	6.94	0.13	2.22	3.49	16.59	0.05	15.67	7.35	8.63	2.56
4			60	6.93	0.13	2.20	3.50	16.78	0.05	16.17	7.42	8.81	2.19
5			baseline	6.77	0.14	2.32	3.14	14.35	0.05	17.93	7.44	8.69	0.10
6		Sample Middle	0	6.53	0.13	2.22	3.19	15.32	0.04	14.95	7.20	8.46	2.64
7			30	6.74	0.19	2.30	3.32	16.40	0.05	15.32	7.37	8.65	2.62
8			60	6.77	0.16	2.31	3.37	16.79	0.05	15.55	7.38	8.67	2.22
9			baseline	7.27	0.25	2.29	4.61	12.17	0.05	17.10	7.13	8.39	0.04
10		Sample Bottom	0	7.04	0.20	2.27	3.43	14.61	0.05	16.00	7.46	8.96	2.00
11			30	7.15	0.29	2.29	3.37	15.64	0.04	15.16	7.38	8.70	2.62
12			60	7.15	0.28	2.32	3.38	15.95	0.04	15.28	7.40	8.61	2.32
13	10 min		baseline	7.49	0.23	2.28	3.44	13.31	0.05	19.08	7.84	9.37	0.02
14		Sample Top	0	6.38	0.35	1.98	2.83	13.10	0.05	17.11	7.78	9.40	1.54
15			30	7.40	0.25	2.27	3.45	16.70	0.05	17.38	7.82	9.39	1.43
16			60	7.40	0.34	2.03	3.38	16.98	0.05	17.35	7.72	9.31	1.14
17		Sample Middle	0	7.35	0.26	2.03	3.16	15.88	0.05	17.38	7.66	8.96	1.49
18			30	7.21	0.29	2.23	3.14	15.96	0.06	17.42	7.72	8.98	1.36
19			60	7.22	0.23	2.12	3.15	16.07	0.06	17.18	7.53	8.85	1.20
20		Sample Bottom	0	7.20	0.29	2.15	3.18	16.22	0.06	17.42	7.42	8.64	1.03
21			30	7.22	0.28	2.16	3.18	16.21	0.06	17.29	7.64	9.01	1.31
22			60	7.14	0.27	2.15	3.15	16.08	0.06	17.15	7.54	8.82	1.21
23	1 min		baseline	6.35	0.07	1.89	3.03	12.43	0.04	13.45	7.50	7.64	0.10
24		Sample Top	0	6.48	0.19	1.96	3.19	16.48	0.06	14.09	7.52	7.99	0.14
25			30	6.15	0.16	1.88	3.07	15.92	0.05	13.76	7.61	7.84	0.24
26			60	6.32	0.17	1.94	3.18	16.42	0.05	13.71	7.64	7.84	0.20
27		Sample Middle	0	6.32	0.12	1.93	3.15	16.33	0.05	13.84	7.55	7.78	0.23
28			30	6.32	0.12	1.94	3.18	16.46	0.05	13.67	7.63	7.78	0.29
29			60	5.50	0.15	1.68	2.65	13.80	0.05	13.91	7.71	7.98	0.20
30		Sample Bottom	0	5.44	0.17	1.70	2.60	13.50	0.05	13.78	7.61	7.87	0.15
31			30	5.64	0.14	1.77	2.72	14.18	0.05	13.82	7.69	7.98	0.18
32			60	6.36	0.20	1.94	3.20	16.54	0.05	13.72	7.62	7.81	0.19

## Data Availability

Data are contained within the article.

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
