# Peer review of "Analyses of the Chemical Composition of Plasma-Activated Water and Its Potential Applications for Vaginal Health"

_biomedicines, 2023, doi:10.3390/biomedicines11123121_

Round 1

Reviewer 1 Report

Comments and Suggestions for Authors

This paper contains useful information. It is worth documented for future researchers’ reference. I only have two minor questions and comments.

1.       Figure 4. It is not clear about what authors intend to demonstrate in Figure .4. No apparent difference is shown in Figure 4 between a and b. Would the authors please provide more descriptions and explanations about Figure 4.

2.       Have the authors test how long the activity or reactivity of PAW last after plasma treatment?

Author Response

Dear Reviewer,

We truly appreciate your useful comments.

We have checked the manuscript according to your comments and marked the corrections by yellow highlight and red color.

In addition, we have attached files such as  revised manuscript (biomedicines-2708293_revision version) and comment response report.

We are resubmitting our manuscript ID ( biomedicines-2708293): Title “Analyses of the Chemical Composition of Plasma-Activated Water and Its Potential Applications for Vaginal Health”

We look forward to your positive response.  If you have any questions, please feel free to contact us.

Thank you so much for your time and consideration. 

Sincerely yours,

Ju-Seop Kang, MD, PhD

Department of Pharmacology and Clinical Pharmacology Lab, College of Medicine, Hanyang University, 17 HaengDang-Dong, SeongDong-Gu, Seoul 133-791, Republic of Korea

Tel: +82-2-2220-0652, Fax: +82-2292-6686, E-mail: jskang@hanyang.ac.kr

Reviewer 2 Report

Comments and Suggestions for Authors

Analysis of Antibacterial Efficacy and Unique Chemical  Compositions Generated from Plasma Activated Water is very interesting paper. Some improvement is required.

Line 16: including ions (which type of ions)

Line 23: primarily HOCl and some cations (Ca2+ and Mg2+), thereby. Is the presence of HOCl dangerous for the health?

Line 184,185: The retention times for individual anions (Fluoride, chloride, Nitrite, Bromide, Nitrate, Phosphate, Sulfate) were approximately 3.74, 5.14, 6.05, 7.06, 7.87, 9.52, and 14.30 minutes, respectively ( Did you calculate these values or measured?)

Line 192: Figure 2 is not visible. Can you offer better visible Figure

Line 232: In tabele you included mg/L and (ppm).  Can you make conversion of ppm in mg/L in Table 3 and Table 4.

Line 242, 242: The presence of HOCl or OCl- depends on the acidity or basicity of the water, with  HOCl being formed under the acidic condition and OCl- being formed under the basic condition (At which pH-value?)

Line 291 , 292: Therefore, the generation of ions, specifically Ca2+ and Mg2+, by PAW through plasma treatment, suggests its potential for antibacterial and  skin protection effects. Why not Cu and Silver Ions? These have many antibacterial effects.

General questions:

What are criteria for the choice of anions and cations in chapter 2.3.1

Why the influence of treatment time on concentration of HOCl is not significant

Author Response

(The authors gave the same response as above.)
